# Reduced miR-146a-5p Is a Biomarker of Infant Respiratory Diseases Contributing to Immune Dysregulation in Small Airway Epithelial Cells

**DOI:** 10.3390/cells11172746

**Published:** 2022-09-02

**Authors:** José M. Rodrigo-Muñoz, Marta Gil-Martínez, Clara Lorente-Sorolla, Beatriz Sastre, María Luz García-García, Cristina Calvo, Inmaculada Casas, Victoria del Pozo

**Affiliations:** 1Department of Immunology, Instituto de Investigación Sanitaria Fundación Jiménez Díaz (IIS-FJD, UAM), 28040 Madrid, Spain; 2Centro de Investigación Biomédica en Red (CIBER) de Enfermedades Respiratorias (CIBERES), 28029 Madrid, Spain; 3Pediatrics Department, Severo Ochoa Hospital, 28911 Leganés, Spain; 4Translational Research Network in Pediatric Infectious Diseases (RITIP), Madrid, Spain; 5Department of Pediatrics, Alfonso X El Sabio University, 28691 Madrid, Spain; 6Centro de Investigación Biomédica en Red (CIBER) de Enfermedades Infecciosas (CIBERINFEC), 28029 Madrid, Spain; 7Pediatric Infectious Diseases Department, Hospital Universitario La Paz, 28046 Madrid, Spain; 8Fundación Instituto de Investigación Hospital Universitario La Paz, IdiPaz, 28029 Madrid, Spain; 9TEDDY Network (European Network of Excellence for Pediatric Clinical Research), 27100 Pavia, Italy; 10Respiratory Virus and Influenza Unit, National Microbiology Center (ISCIII), 28220 Madrid, Spain

**Keywords:** infant respiratory diseases, microRNAs, immune regulation

## Abstract

Respiratory diseases such as bronchiolitis, and those with wheezing episodes, are highly important during infancy due to their potential chronicity. Immune response dysregulation is critical in perpetuating lung damage. Epigenetic modifications including microRNA (miRNA) post-transcriptional regulation are among the factors involved in alleviating inflammation. We evaluated the expression of miR-146a-5p, a previously described negative regulator of immunity, in infants with respiratory diseases, in order to study epigenetic regulation of the immune response. Nasopharyngeal aspirate (NPA) was obtained from infants with bronchiolitis (ongoing and post-disease) or with wheezing episodes in addition to healthy controls. Virus presence was determined by nested PCR, while miRNA and gene expression were studied in cells from NPAs using qPCR. Healthy small airway epithelial cells (SAECs) were used as an in vitro model. We observe a reduction in miR-146a-5p expression in infants with either of the two diseases compared to controls, suggesting the potential of this miRNA as a disease biomarker. Post-bronchiolitis, miR-146a-5p expression increases, though without reaching levels of healthy controls. MiR-146a-5p expression correlates inversely with the immune-related gene *PTGS2*, while its expression correlates directly with *TSLP*. When heathy donor SAECs are stimulated by poly:IC, we observe an increase in miR-146a-5p, with wounds having a synergistic effect. In conclusion, infants with respiratory diseases present reduced miR-146a-5p expression, possibly affecting immune dysregulation.

## 1. Introduction

Infant respiratory diseases are a worldwide health issue due to their increased prevalence and worsening outcomes and severity [1], which contribute to chronic respiratory morbidity, including asthma [2]. Bronchiolitis is the most frequent cause of lower respiratory tract infections, leading to hospitalizations at a young age [3]. Features of the disease in patients under age two years comprise a series of clinical symptoms including viral upper respiratory prodrome, respiratory effort, and wheezing [4]. Around 60% to 80% of all cases of bronchiolitis are caused by respiratory syncytial virus (RSV) [5], while human rhinovirus (HRV), metapneumovirus, bocavirus, and parainfluenza virus are also involved, though to a lesser extent [6]. Viral lung infections and bronchiolitis were previously associated with the development of wheezing and asthma symptoms at age six to eight years [7,8], and preterm birth was found to be an important risk factor for this association [9]. Preschool wheezing is a highly prevalent sign of bronchiolitis, affecting around one-third of infants younger than five years of age [10]; although it can resolve by school age without intervention, some infants with wheezing develop chronic asthma [10]. 

Both bronchiolitis and wheezing episodes are characterized by a dysregulation of the immune response and have shared mechanisms, such as vascular damage, fibroblast repair, and monocyte activation; although the two diseases are related, there are differing degrees of increase in the innate immune response [11]. Studies found that around one-fifth of subjects with bronchiolitis later develop wheezing episodes [12]. Although these diseases are very common, their high prevalence has yet to be explained. Some studies described risk factors including family history of asthma, low birth weight, and/or exposure to toxic substances [13]. 

Individual predisposition to bronchiolitis or wheezing episodes may be explained by analyzing genetic information. A genome-wide association study reports polymorphisms possibly associated with bronchiolitis [14], while other research found genetic polymorphisms related to respiratory morbidity in preterm infants with RSV infection, linking bronchiolitis to symptoms of lung dysfunction [15]. Alternatively, the predisposition toward immune dysregulation in infants with these respiratory diseases may be explained by epigenetic modifications such as microRNAs (miRNAs), which are small non-coding RNAs that perform post-transcriptional regulation, making them promising candidates as disease biomarkers [16]. 

The aim of this study is to analyze the expression of miR-146a-5p, an miRNA previously related to the fine-tuning of the immune response due to its ability to prevent overstimulation of the inflammatory response [17]. To determine the role of miR-146a-5p in infant respiratory diseases, we compared levels of miR-146a-5p expression in nasopharyngeal samples from infants with bronchiolitis (both ongoing disease and after disease remission) and infants with wheezing episode versus a control group consisting of healthy infants.

## 2. Materials and Methods

### 2.1. Study Design

This prospective, cross-sectional, single-site study was conducted at the Severo Ochoa University Hospital (Leganés, Madrid, Spain) starting in October 2016. Participating infants belonged to any of three categories: healthy controls (disease-free infants recruited during admission for elective surgery or food challenge); hospitalized infants with bronchiolitis, 12 of whom provided an additional sample after discharge (this sample was labeled as post-bronchiolitis, defined as a negative viral determination and lack of symptoms after a recovery period); and hospitalized infants with an episode of wheezing, who were subdivided into virus-positive and virus-negative wheezing episode in order to detect miR-146a-5p differences related to viral status. Parents of participating subjects completed a questionnaire to rule out presence of respiratory symptoms over the preceding 10 days (e.g., cough, fever > 38 °C, sore throat, rhinitis, acute otitis media, pneumonia, wheezy bronchitis). Inclusion criteria for bronchiolitis were all infants younger than 24 months admitted with a diagnosis of this disease as defined by McConnochie, i.e., the first episode of acute-onset expiratory dyspnea with previous signs of viral respiratory infection [18]. A wheezing episode was defined as the presence of expiratory dyspnea and wheezing diagnosed by a physician in children with a previous episode of bronchiolitis, and being admitted with the presence of hypoxia, apnea, respiratory distress, and refusal of feeding [7]. These patients were ruled out for bronchiolitis group, as these were no more than their first episode of acute-onset expiratory dyspnea with previous signs of viral respiratory infection. Signed informed consent was obtained from parents or legal guardians. During their hospital stay, children were evaluated clinically by a physician, who prospectively completed a structured questionnaire on symptoms and other clinical variables. The study protocol was approved by the hospital ethics committee, and was conducted in accordance with the principles set forth in the Declaration of Helsinki.

### 2.2. Sample Collection and Virus Detection

Two nasopharyngeal aspirates (NPAs) were obtained by washing the nasal cavity with 1 mL of phosphate-buffered saline; these were then collected using a standard mucus extractor. The samples were refrigerated at 4 °C until they were processed, within 24 h of collection. For clinical purpose, part of each NPA sample was used to detect the sixteen most common respiratory viruses by means of three RT-nested PCR assays in the Respiratory Virus and Influenza Unit at the National Microbiology Center (ISCIII, Madrid, Spain): respiratory syncytial virus (RSV) A and B types, human rhinovirus (HRV), human metapneumovirus (hMPV), human bocavirus (HBoV), adenovirus (AdV), influenza A, B, and C viruses, parainfluenza viruses 1 to 4 (PIV), human coronaviruses (CoV) 229E and OC43, and enteroviruses (EV).

### 2.3. Nasopharyngeal Aspirate (NPA) Processing

The remaining specimen in each NPA sample was filtered with a 40 µm nylon filter and centrifuged to obtain the cellular pellet and supernatant. Following a cell count, pellets were resuspended in 0.7 mL of Qiazol Lysis Reagent (Qiagen, Hilden, Germany) and frozen at –80 °C. Supernatants were directly frozen at –80 °C. 

### 2.4. Small Airway Epithelial Cell Culture

Primary small airway epithelial cells (SAECs) from healthy subjects (Lonza, Basel, Switzerland) were cultured in specific growth medium (Promocell, Heidelberg, Germany). Media were supplemented with 100 U/mL penicillin and 100 µg/mL streptomycin and maintained at 37 °C in an atmosphere containing 5% CO_2_. In order to study the effect of virus infection in the in vitro model, cells were incubated with poly:IC (Sigma Aldrich, St. Louis, MO, USA) that simulated a viral stimulus by using the same cell receptor. Moreover, the role of airway epithelial damage was also assessed in this SAEC in vitro model, and for this purpose, epithelial barrier integrity was disrupted using a pipette tip.

### 2.5. Gene Expression Assays

RNA from the cellular fraction of NPAs and SAECs was obtained using the phenol-chloroform technique from Qiazol (Qiagen) (PMID: 8747660). Five hundred ng of RNA quantified by Nanodrop ND-1000 spectrophotometer (Thermo Fisher Scientific, Waltham, MA, USA) was reverse-transcribed with a High-Capacity cDNA Reverse Transcription Kit (Applied Biosystems, Foster City, CA, USA) and analyzed by semi-quantitative real-time PCR (qPCR) on a 7500 real-time PCR system (Applied Biosystems) with TaqMan^TM^ gene expression probes (Applied Biosystems) for *18s*, *TSLP*, *IFNG*, *TLR3*, *PTGS2*, *NFKB*, *IL1R1*, *IL10*, *POSTN*, and *AREG* mixed with TaqMan^®^ Gene Expression MasterMix (Applied Biosystems). Relative gene expression was calculated using the cycle threshold (Ct) and the 2^−ΔΔCt^ method [19], where
ΔΔCt = ΔCt_population1_ − ΔCt_population2_ and ΔCt = ΔCt_gene_ − ΔCt_Housekeeping gene_

### 2.6. MicroRNA Analysis in NPA Samples

Ten nanograms of NPA RNA were retrotranscribed using the miRCURY LNATM RT Kit (Qiagen) followed by qPCR in a Light Cycler 96 thermocycler (Roche, Basel, Switzerland) using the miRCURY LNA^TM^ SYBR Green PCR Kit and miR-146a-5p, miR-103a-3p, miR-191-5p, RNU6, and UniSp6 probes (Qiagen). The average between data from miR-191-5p, miR-103a-3p, and RNU6 was used as a housekeeping miRNA. Relative miRNA expression was calculated using the Ct and the 2^−ΔΔCt^ method [19]: ΔΔCt = ΔCt_population1_ − ΔCt_population2_ and ΔCt = ΔCt_miRNA_ − ΔCt_Housekeeping miRNA average_.

### 2.7. Statistical Analysis 

Data were compared by unpaired, two-tailed Student’s t test or Mann–Whitney U test for non-Gaussian samples; multiple comparisons were performed with the use of ANOVA with Bonferroni post hoc test or Kruskal–Wallis with Dunn’s multiple comparison. Results are shown as mean ± SEM. *p* values < 0.05 were considered statistically significant. Correlations were estimated by Pearson r or Spearman’s ρ. The performance of miRNAs as a biomarker was evaluated by receiver operating curve (ROC). Statistical calculations were performed and graphs were created with Graph-Pad Prism 6–8 (GraphPad Software Inc., RRID:SCR_002798, San Diego, CA, USA).

## 3. Results

The study population consisted of 10 healthy controls, 41 infants diagnosed with bronchiolitis (of whom 12 infants provided a post bronchiolitis sample), and 20 subjects with wheezing episodes (ten with and ten without viral infection). The clinical characteristics of the study subjects are shown in Table 1. 

### 3.1. MiR-146a-5p Is Reduced in NPAs from Infants with Respiratory Diseases and Correlates with Immune-Related Genes

We observed that miR-146a-5p expression is reduced in infants with respiratory disease compared to controls (0.18 ± 0.03 vs. 0.85 ± 0.21; *p* < 0.0001, Figure 1A). When patients were subdivided according by diagnosis, we found increased miR-146a-5p expression in NPAs from control infants (0.85 ± 0.21) compared to infants with bronchiolitis (0.16 ± 0.03; *p* < 0.05), virus-positive patients with wheezing episodes (0.15 ± 0.05; *p* < 0.01), and with respect to the virus-negative, wheezing episode group (0.26 ± 0.09; *p* < 0.0001), as seen in Figure 1B. Of note, the ROC curve analysis shows that miR-146a-5p expression in NPAs could be used as a biomarker for infant respiratory disease (AUC = 0.85; 95% CI = 0.71–0.98; *p* < 0.05, Figure 1C). 

To understand how the immune dysregulation of these patients is associated with the expression of this miRNA, we correlated miR-146a-5p expression with the prostaglandin endoperoxide synthase 2 (*PTGS2*) gene, also known as cyclooxygenase 2 (COX-2), which is described as a target of miR-146a-5p; an inverse correlation is found (Spearman ρ = −0.43; *p* < 0.05; Figure 2A). We also studied the relationship with other genes related to immune responses (*TSLP*, *NFKB*, *TLR3*, and *IL1R1*) and we observe a direct correlation of miR-146a-5p expression with thymic stromal lymphopoietin (*TSLP*) gene expression (Spearman ρ = 0.35; *p* < 0.05; Figure 2). We do not find any significant correlation between miR-146a-5p expression and the expression of *NFKB*, *TLR3*, or *IL1R1* (Spearman ρ near 0; *p* > 0.05). 

Moreover, we tested if any continuous value of clinical values correlate with miR-146a-5p expression, and there is no correlation between this miRNA expression and age or length of hospital stay (Spearman ρ near 0; *p* > 0.05).

To determine whether miR-146a-5p levels normalize after remission of the disease, we compared the expression in NPAs from infants with ongoing bronchiolitis and those in the post-bronchiolitis phase (defined as negative viral determination and lack of symptoms after a recovery period), observing higher expression of miR-146a-5p after disease remission (2.97 ± 0.06-fold; *p* < 0.05) than during ongoing bronchiolitis, though both are still lower compared to the healthy controls (8.62 ± 1.84-fold; *p* < 0.05; Figure 3). 

### 3.2. Viral Stimuli Increase miR-146a-5p Expression in Small Airway Epithelial Cells from Controls

Aiming to confirm that viral infection alters miR-146a-5p expression, we took an in vitro experimental approach using healthy lung SAECs, which we stimulated with poly:IC, a virus analog. Stimulating SAECs with poly:IC at increasing concentrations (5, 10, and 50 µg/mL) for 3, 6, and 24 h, we observe that the expression of miR-146a-5p increases in a dose-dependent manner, starting at 6 h, increasing at 24 h, and reaching a peak of 12–20 times miR-146a-5p over-expression compared to untreated controls (Figure 4A). 

MiR-146a-5p up-regulation is accompanied by an increase in virus or poly:IC receptor expression (TLR3), which peaks at around 30-fold expression at 24 h, and also by a rise in the production of thymic stromal lymphopoietin (TSLP), an alarmin produced by activated epithelial cells. Maximum production occurs at 6 h (Figure 4B). 

To observe whether damaged epithelium alters the responses of miR-146a-5p expression in SAEC, we repeated the former experiment, this time creating wounds in the cell monolayer with a pipette tip, using poly:IC (20 ng/mL), or both. Poly:IC increases miR-146a-5p expression (3.4 ± 0.5-fold expression; *p* > 0.05), though the wounds alone do not alter miR-146a-5p levels (0.8 ± 0.2). However, wounded cells treated with poly:IC show significantly increased miR-146a-5p expression (11.2 ± 5.1-fold expression; *p* < 0.05) to the highest levels, suggesting a possible synergistic effect between virus infection and epithelial damage in miR-146a-5p up-regulation (Figure 5A). As previously, both poly:IC alone or with wounds increases TLR3 (*p* < 0.05) and TSLP expression (*p* > 0.05) in SAECs, which is evidence that the alteration is caused by viral stimuli and epithelial damage in the immune response (Figure 5B). Nevertheless, unlike with miR-146a-5p expression, there is no synergistic effect on TLR3 and TSLP expression caused by poly:IC plus wounds.

## 4. Discussion

Our results show reduced miR-146a-5p expression in infants with lower respiratory bronchiolitis and others experiencing wheezing episodes. MiR-146a-5p is a miRNA that shows increased expression in some viral and inflammatory environments, and which acts as a negative immune regulator, thus, controlling the resolution of inflammation [17].

Epigenetic regulation, including miRNA post-transcriptional repression, has been studied extensively over recent decades, due to its role in fine-tuning immune responses [20]. Of these, miR-146a-5p has been particularly prominent, as several studies addressed the behavior of this miRNA and the control it exerts in a range of cellular models. It has been reported to be increased both by pathogens such as coxsackievirus B (CVB) [21] or hepatitis C virus [22], as well as by pathogen-related molecules such as endotoxins (LPS) [23]. When stimulated by pathogen and pro-inflammatory mediators, miR-146a-5p performs its main action, which relies on its capacity to act as a negative feedback loop of immune response control, thus, enabling this miRNA to suppress the NF-κβ pathway [21]. Interestingly, we observe a negative association between *NFKB* and miR-146a-5p in NPAs from bronchiolitis, showing an increase in inflammatory responses accompanied by a decrease in miR-146a-5p, suggesting that immune responses are not correctly resolved. Moreover, the action performed by miR-146a-5p in anti-inflammatory control includes regulation of IL-1β, as this molecule itself activates miR-146a-5p expression; this miRNA then causes IL-1β downstream signals to shut down, as it reduces IL-6, TNF-α, and COX2 [24]. Indeed, cyclooxigenase-2 (COX2, encoded by the gene *PTGS2*) is an important inflammatory mediator in several diseases, and is a target for miR-146a-5p described in bone-marrow-derived mesenchymal stem cells (BMSCs) [25]. We also observe an inverse correlation of both *IL1R1* and *PTGS2* with miR-146a-5p gene expression in bronchiolitis NPAs, suggesting that the reduction in this miRNA could be related to a failure of inflammation resolution.

MiR-146a-5p has been also studied in chronic respiratory diseases such as asthma, a disease where wheezing is a key sign: this miRNA is found to be reduced in bronchial brushings from asthmatic individuals, and when over-expressed in epithelial cells, it reduces IL-8 and CXCL1 [26]. Interestingly, bronchial smooth muscle cells from asthmatics exhibit higher miR-146a-5p expression in response to IL-1β, TNF-α, and IFNγ compared to cells from controls, showing intercellular differences in miRNA responses [27]. Biomarker studies in body fluids of asthmatics have also proven differential expression of this miRNA, which in increased in plasma from asthmatic adults [28], in plasma from asthmatic children [29], and in serum from allergic asthmatics [30]. The discrepancies between miR-146a-5p expression in plasma (increased in asthma) and epithelial cells (increased by TNF-α and reduced by corticosteroids) were further discussed by Lambert et al., whose results highlight the anti-inflammatory negative control loop performed by this miRNA, and how it should be over-expressed in asthma to enhance treatment efficacy [31]. The link between miR-146a-5p, viral infections, and asthma was developed in mouse models, in which RSV increased miR-146a-5p expression, while miR-146a-5p knockout mice developed higher asthma exacerbations when infected with RSV, thereby highlighting the anti-inflammatory role of this miRNA in asthma [32].

Although many studies, including the present one, show that a viral stimulus enhances miR-146a-5p expression as an inflammatory control mechanism in epithelial cells [33], this mechanism seems to become abnormal in some disease conditions. First, A549 cells infected with RSV show diminished miR-146a-5p in their exosomes, indicating a reduction in miRNA secreted for intercellular communication in the context of infection [34]. More importantly, and in accordance with our results, RSV infection reduces miR-146a-5p expression in A549, HEp-2 cells, and in rat lung tissues; and over-expression of miR-146a-5p in the animal model reduces inflammation by inhibiting the molecular pathways related to NF-κβ activation induced by RSV [17,35]. Similarly, although our viral stimulus (poly:IC) causes miR-146a-5p up-regulation, as previously described in airway epithelial cells in vitro [35], our patients with viral respiratory infections and wheezing episodes have reduced miR-146a-5p expression. These results clearly demonstrate the complex mechanisms of immunity, and support our finding that reduced levels of miR-146a-5p in infant respiratory diseases can be related to chronic inflammation and respiratory symptoms after bronchiolitis, even in those infants who have overcome the period of active infection, as these patients tend to have increased miR-146a-5p levels compared to those with ongoing viral infection, though these levels do not reach those of controls. Moreover, infants with bronchiolitis and wheezing episodes have reduced miR-146a-5p expression levels compared to healthy controls, which suggests that expression of this miRNA in airway samples from infants could be a useful biomarker for determining which patients may present immune response dysregulation and consequent failure to resolve inflammation.

## 5. Conclusions

In summary, our results show that miR-146a-5p is reduced in the airways of infants with severe lower respiratory bronchiolitis and with wheezing episodes, suggesting the suitability of this miRNA as a biomarker for the disease, as well as a measure of immune dysregulation in small airway epithelial cells in vitro. Artificial over-expression of miR-146a-5p, such as that observed in many animal models, may, thus, be a promising therapeutic target to restore immunity and resolve inflammation in infants with severe early respiratory infections and wheezing. 

## Figures and Tables

**Figure 1 cells-11-02746-f001:**
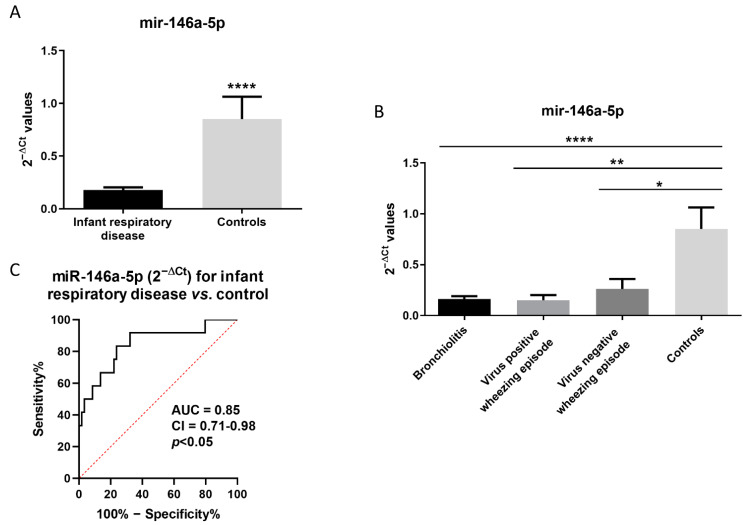
miR-146a-5p is altered in NPA samples from infant respiratory diseases. (**A**) miR-146a-5p expression (2^−ΔCt^) in infant subjects with respiratory disease and controls. (**B**) miR-146a-5p expression (2^−ΔCt^) in patients with bronchiolitis, virus-positive patients with wheezing episodes, virus-negative subjects with wheezing episodes, and controls. (**C**) ROC curve for miR-146a-5p expression (2^−ΔCt^) as a biomarker for infant respiratory diseases. AUC = area under the curve; CI = confidence interval. * *p* < 0.05; ** *p* < 0.01; **** *p* < 0.0001.

**Figure 2 cells-11-02746-f002:**
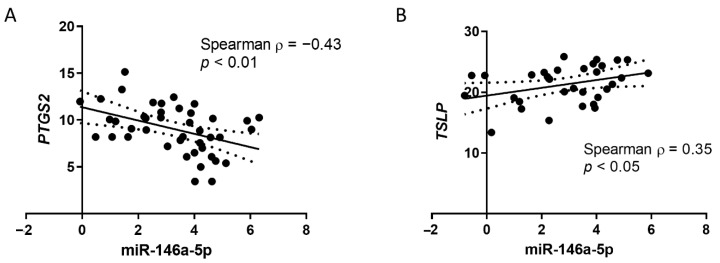
miR-146a-5p correlates with genes involved in immune responses. (**A**) Gene expression correlation (2^−ΔCt^) of miR-146a-5p and PTGS2 in NPA cells from infants with respiratory diseases. (**B**) Gene expression correlation (2^−ΔCt^) of miR-146a-5p and TSLP in NPA cells from infants with respiratory diseases. Lines indicate linear regression and dotted lines indicate 95% confidence bands of the best-fit line.

**Figure 3 cells-11-02746-f003:**
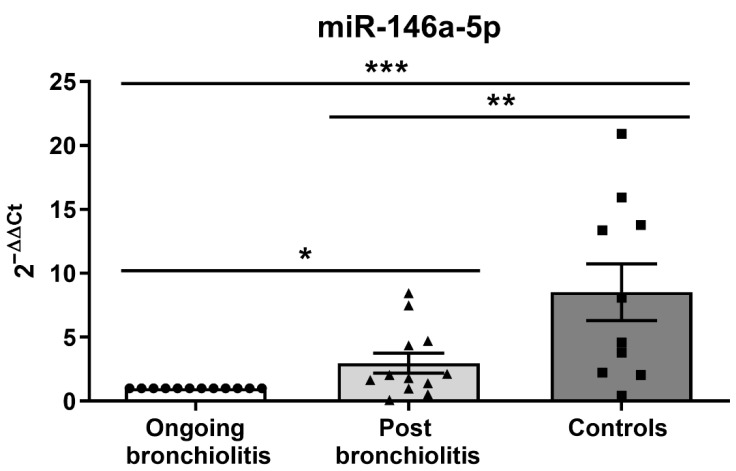
miR-146a-5p levels in NPA samples of patients with ongoing bronchiolitis and patients in post-recovery. MiR-146a-5p expression (2^−ΔΔCt^) in infants with ongoing bronchiolitis, post-bronchiolitis, and controls. * *p* < 0.05; ** *p* < 0.01; *** *p* < 0.001. Dots represent ongoing bronchiolitis individuals; triangles represent post bronchiolitis individuals and squares represent control individuals.

**Figure 4 cells-11-02746-f004:**
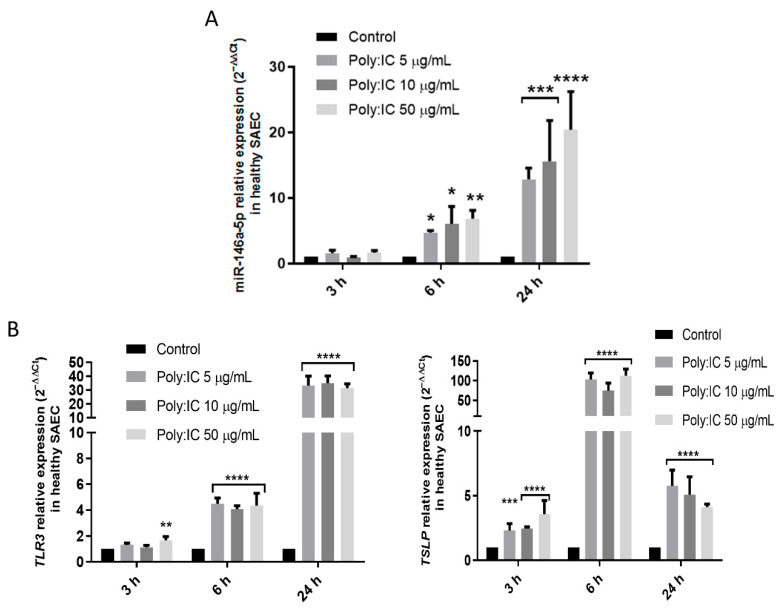
miR-146a-5p levels are increased when exposed to viral analog stimuli in healthy airway epithelial cells. (**A**) miR-146a-5p expression (2^−ΔΔCt^) increases in SAECs after stimulation with poly:IC in increasing doses (5, 10, and 50 µg/mL). (**B**) TLR3 and TSLP expression (2^−ΔΔCt^) are augmented in SAEC after stimulation with poly:IC in increasing doses (5, 10, and 50 µg/mL). SAEC: Small airway epithelial cells. * *p* < 0.05; ** *p* < 0.01; *** *p* < 0.001; **** *p* < 0.0001 (compared to the untreated control group).

**Figure 5 cells-11-02746-f005:**
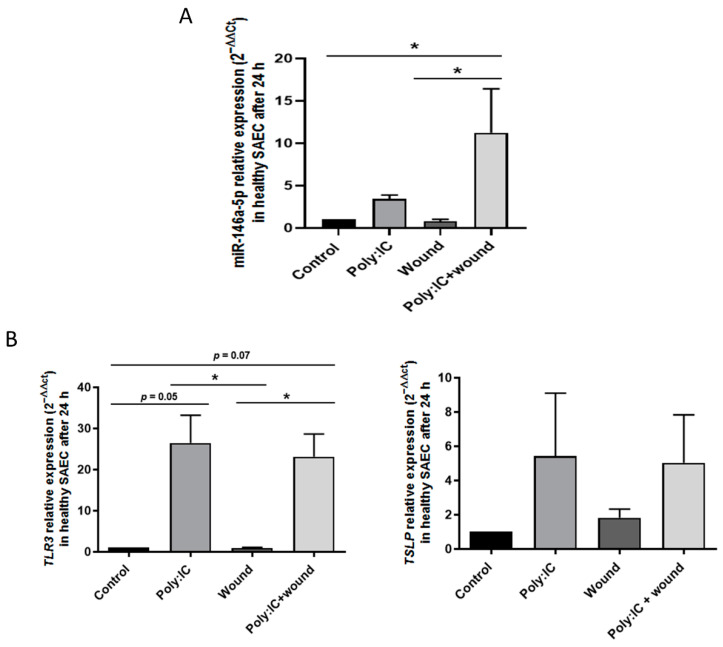
Poly:IC and epithelial wounds produce a synergistic increase in miR-146a-5p (**A**) miR-146a-5p expression (2^−ΔΔCt^) in SAECs influenced by the action of poly:IC (20 ng/mL), epithelial wounds, and combination of both. (**B**) TLR3 and TSLP expression (2^−ΔΔCt^) are altered in SAECs after stimulation with poly:IC (20 ng/mL) and poly:IC plus wounds. SAEC: Small airway epithelial cells. * *p* < 0.05.

**Table 1 cells-11-02746-t001:** Demographic and clinical data of study subjects.

	Controls (*n* = 10)	Bronchiolitis (*n* = 41)	Viral-Positive, Wheezing Episode (*n* = 10)	Viral-Negative, Wheezing Episode (*n* = 10)
Age (mo)	4.9 ± 0.9	3.5 ± 0.4	33.3 ± 11.1	16.9 ± 3.8
Male (%)	60.0 (6/10)	51.2 (21/41)	77.8 (7/9)	40.0 (4/10)
Prematurity (%)	NA	4.1 (1/24)	11.1 (1/9)	0.0 (0/10)
Hospital stay (d)	NA	5.1 ± 0.6	2.6 ± 0.5	2.7 ± 0.5
Temperature > 37.9 °C (%)	NA	12.5 (3/24)	55.6 (5/9)	33.3 (3/9)
Hypoxia (SatO2 < 95%) (%)	NA	79.2 (19/24)	88.8 (8/9)	55.5 (5/9)
Neonatal admission (%)	NA	0.0 (0/24)	11.1 (1/9)	0.0 (0/9)
Neonatal CPAP (%)	NA	0.0 (0/22)	11.1 (1/9)	0.0 (0/9)
Antibiotic treatment (%)	NA	8.3 (2/24)	11.1 (1/9)	22.2 (2/9)
Virus (%)	NA	95.1 (39/41)	100.0 (10/10)	0.0 (0/10)
RSV	NA	58.5 (24/41)	50.0 (5/10)	NA
HRV	NA	34.1 (14/41)	60.0 (6/19)	NA
Coinfection	NA	24.4 (10/41)	20.0 (2/19)	NA

Mo: months; SatO2: oxygen saturation; CPAP: continuous positive airway pressure; RSV: respiratory syncytial virus; HRV: human rhinovirus; NA: not available.

## Data Availability

The data that support the findings of this study are available from the corresponding author, VdP, upon reasonable request.

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
