# Peer review of "Reduced miR-146a-5p Is a Biomarker of Infant Respiratory Diseases Contributing to Immune Dysregulation in Small Airway Epithelial Cells"

_cells, 2022, doi:10.3390/cells11172746_

Round 1
Reviewer 1 Report
Dear Authors,
I read with interest your article entitled "Reduced miR-146a-5p is a biomarker of infant respiratory dis-2 eases contributing to immune dysregulation”. The protocol is well designed, and the article is well written. The limitation of the study is the low number of children examined
Please go through my comments:
- As there are intracellular differences in miRNA responses (mechanism of miR-146a function is cell-type dependent) and examined aspect concern epithelial cells only, this should be mentioned in the title as well as in the conclusion (lines 311-312).
- Lines 53-56: I don’t agree that mentioned diseases trigger an increase in the immune response – they are related with.
- Lines 85-88 vs. 153-155: you should be precise to whom lines 85-88 are addressed
- Children with bronchiolitis had the lowest age: is the miR-146a-5p expression age-related? This aspect should be discussed.
Author Response
Reviewer 1
Dear Authors,
I read with interest your article entitled "Reduced miR-146a-5p is a biomarker of infant respiratory diseases contributing to immune dysregulation”. The protocol is well designed, and the article is well written. The limitation of the study is the low number of children examined
Please go through my comments:
POINT 1
- As there are intracellular differences in miRNA responses (mechanism of miR-146a function is cell-type dependent) and examined aspect concern epithelial cells only, this should be mentioned in the title as well as in the conclusion (lines 311-312).
RESPONSE 1: We agree with the reviewer comment. We have modified the title to: “Reduced miR-146a-5p is a biomarker of infant respiratory diseases contributing to immune dysregulation in small airway epithelial cells”. Furthermore, we have addressed this same issue in the conclusion stating that “In summary, our results show that miR-146a-5p is reduced in the airways of infants with severe lower respiratory bronchiolitis and recurrent wheezing, suggesting the suitability of this miRNA as a biomarker for the disease and as well as a measure of immune dysregulation in small airway epithelial cells in vitro.” As seen in lines 323 at page 10.
POINT 2
- Lines 53-56: I don’t agree that mentioned diseases trigger an increase in the immune response – they are related with.
RESPONSE 2: Thank you for the correction, we have changed “trigger” to “are related with” in line 57 at page 2.
POINT 3
- Lines 85-88 vs. 153-155: you should be precise to whom lines 85-88 are addressed
RESPONSE 3: We have changed this lack of information stating that “hospitalized infants with bronchiolitis, from whom 12 of them provided an additional sample after discharge (this sample was labeled as post-bronchiolitis, defined as a negative viral determination and lack of symptoms after a recovery period)” as depicted in line 85 at page 2, and we have also described this information saying that “The study population consisted of ten healthy controls, 41 infants diagnosed with bronchiolitis (of whom 12 infants had a post bronchiolitis sample), and 20 subjects with recurrent wheezing (ten with and ten without viral infection).” As seen in line 164-167 at page 4.
POINT 4
- Children with bronchiolitis had the lowest age: is the miR-146a-5p expression age-related? This aspect should be discussed.
RESPONSE 4: Indeed, we have tested if age had any correlation with the expression of miR-146a-p, but there was no correlation with it (Spearman r= -0.11, p>0.05). We stated this information saying that “Moreover, we tested if any continuous value of clinical values correlated with miR-146a-5p expression, not being any correlation between this miRNA expression and age or length of hospital stay (Spearman ρ near 0; p>0.05).” as seen in lines 199-201 at page 6.
Reviewer 2 Report
This study aims to investigate the expression of miR-146a-5p, a miRNA able to prevent overstimulation of the inflammatory response, in infants with bronchiolitis or recurrent wheezing. To this aim, nasopharyngeal aspirates were collected from 3 groups of infants: 41 with bronchiolitis, 20 with recurrent wheezing and 10 healthy controls. The expression of miR-146a-5p was reduced in infants with respiratory diseases and correlated negatively with the gene expression of PTGS2 and positively with gene expression of TSLP. In vitro analysis of healthy small airway epithelial cells stimulated with different doses of poly:IC shows a dose-dependent increased of miR-146a-5p expression. This increase was paralleled by increased gene expression of the virus or poly:IC receptor TLR3 and of the alarmin TSLP. To observe whether miR-146a-5p expression was affected by epithelium damage, in vitro experiments were repeated creating wounds in the cell monolayer. The baseline expression of miR-146a-5p in the wounded monolayer was similar to that in the intact monolayer. However, the increase of miR-146a-5p expression in response to poly:IC was greater in the wounded monolayer than in the intact monolayer, suggesting a synergistic effect. The authors concluded that miR-146a-5p expression was reduced in infants with respiratory bronchiolitis and recurrent wheezing.
There are some concerns with this manuscript
1) The paragraph study design as well as the paragraph at the end of the introduction report that 3 groups of infants were recruited, however table 1 includes the clinical characteristics of 4 group of subjects. Why recurrent wheezing subjects were divided in two groups? This should be clearly explained.
2) It is not described how recurrent wheezing was diagnosed. Patients included in reference 19 were “children 6–8 years of current age, with a previous hospitalization with bronchiolitis at 0–24 months of age”. They could not be defined “with recurrent wheezing”. Furthermore, the paragraph “study design” and table 1 refers to “hospital stay”. Why and when recurrent wheezing infants have been hospitalized?
3) In general, table 1 includes a lot of data that seems poorly related to the aim and the design of the present study. Indeed the majority of this data are not quoted along the manuscript. Possibly, some of them could be deleted. Was there any correlation between miR-146a-5p expression and the clinical characteristics?
4) Paragraph 2.2: Which respiratory virus were examined and why?
5) Paragraph 2.4 should briefly explain the reason for incubate cells with poly:IC and for disrupting epithelial barrier integrity
6) Paragraph 3.1 “We did not find any significant correlation between miR-146a-5p expression and expression of NFKB, TLR3 or IL1R1” Different results were reported in the abstract: “miR-146a-5p expression correlated inversely with NFKB and IL1R1”. Moreover, all p value in the text are <0.05 while in figure 1A and 1B p values ranges from <0.05 to <0.0001.
7) Figure 3: it should be of interest to present the results showing the individual value during bronchiolitis related to the individual value after disease remission in each case. How many of the 41 infants with bronchiolitis have the additional sample after disease remission?
8) Figure 4: it is difficult to understand to which comparisons the asterisks refer.
9) Figure 5B: poly:IC stimulation increased TLR3 and TSLP expression but the effect in the wounded monolayer is the same as in the intact monolayer. There is no synergistic effects of poly:IC stimulation and epithelial damage.
Author Response
Reviewer 2
Comments and Suggestions for Authors
This study aims to investigate the expression of miR-146a-5p, a miRNA able to prevent overstimulation of the inflammatory response, in infants with bronchiolitis or recurrent wheezing. To this aim, nasopharyngeal aspirates were collected from 3 groups of infants: 41 with bronchiolitis, 20 with recurrent wheezing and 10 healthy controls. The expression of miR-146a-5p was reduced in infants with respiratory diseases and correlated negatively with the gene expression of PTGS2 and positively with gene expression of TSLP. In vitro analysis of healthy small airway epithelial cells stimulated with different doses of poly:IC shows a dose-dependent increased of miR-146a-5p expression. This increase was paralleled by increased gene expression of the virus or poly:IC receptor TLR3 and of the alarmin TSLP. To observe whether miR-146a-5p expression was affected by epithelium damage, in vitro experiments were repeated creating wounds in the cell monolayer. The baseline expression of miR-146a-5p in the wounded monolayer was similar to that in the intact monolayer. However, the increase of miR-146a-5p expression in response to poly:IC was greater in the wounded monolayer than in the intact monolayer, suggesting a synergistic effect. The authors concluded that miR-146a-5p expression was reduced in infants with respiratory bronchiolitis and recurrent wheezing.
There are some concerns with this manuscript
POINT 1
The paragraph study design as well as the paragraph at the end of the introduction report that 3 groups of infants were recruited, however table 1 includes the clinical characteristics of 4 group of subjects. Why recurrent wheezing subjects were divided in two groups? This should be clearly explained.
REPONSE 1: We agree with the reviewer that this might be unclear, that is why we have clarified that:“hospitalized infants with an episode of recurrent wheezing were subdivided into virus positive and virus negative recurrent wheezing in order to detect miR-146a-5p differences related to viral status.” As observed in lines 88-90 at page 2.
POINT 2
It is not described how recurrent wheezing was diagnosed. Patients included in reference 19 were “children 6–8 years of current age, with a previous hospitalization with bronchiolitis at 0–24 months of age”. They could not be defined “with recurrent wheezing”. Furthermore, the paragraph “study design” and table 1 refers to “hospital stay”. Why and when recurrent wheezing infants have been hospitalized?
RESPONSE 2: We used the definition of recurrent wheezing included in the reference 19 “Role of viral coinfections in asthma development.”. Recurrent wheezing was defined as more than one episode of wheezing verified by a physician. All children not belonging to the control group, bronchiolitis and recurrent wheezing, were hospitalized for their respiratory pathology and were recruited during admission. Admission criteria were the usual in clinical practice: hypoxia, apnea, respiratory distress, refusal of feeding. We have added some of this information for clarifying in lines 96-97 at page 2.
POINT 3
In general, table 1 includes a lot of data that seems poorly related to the aim and the design of the present study. Indeed the majority of this data are not quoted along the manuscript. Possibly, some of them could be deleted. Was there any correlation between miR-146a-5p expression and the clinical characteristics?
RESPONSE 3: We added most if this information in order to provide the readers with as much information as possible regarding the study groups. We correlated miR-146a-5p expression with clinical parameters with continuous values, not being any clear correlation, as now depicted in the manuscript: “Moreover, we tested if any continuous value of clinical values correlated with miR-146a-5p expression, not being any correlation between this miRNA expression and age or length of stay (Spearman ρ near 0; p>0.05).” as seen in lines 199-201 at page 6.
POINT 4
Paragraph 2.2: Which respiratory virus were examined and why?
RESPONSE 4: Regarding this, three reverse transcription (RT)-nested PCR assays were performed to detect the sixteen more frequent respiratory viruses: respiratory syncytial virus (RSV) A and B types, human rhinovirus (HRV), human metapneumovirus (hMPV), human bocavirus (HBoV), adenovirus (AdV), influenza A, B and C viruses, parainfluenza viruses 1 to 4 (PIV), human coronaviruses (CoV) 229E and OC43 and enteroviruses (EV). We screened respiratory viruses mainly for clinical purposes. We added this information in the manuscript lines 110-114 at page 3.
POINT 5
Paragraph 2.4 should briefly explain the reason for incubate cells with poly:IC and for disrupting epithelial barrier integrity
RESPONSE 5: We have added the research objective of this experiments in the manuscript by writing that “In order to study the effect of virus infection in the in vitro model, cells were incubated with poly:IC (Sigma Aldrich, MO, USA) that simulate a viral stimulus by using the same cell receptor. Moreover, the role of airway epithelial damage was also assessed in this SAEC in vitro model, and for this purpose, epithelial barrier integrity was disrupted using a pipette tip.” As seen in lines 126-129 at page 3.
POINT 6
Paragraph 3.1 “We did not find any significant correlation between miR-146a-5p expression and expression of NFKB, TLR3 or IL1R1” Different results were reported in the abstract: “miR-146a-5p expression correlated inversely with NFKB and IL1R1”. Moreover, all p value in the text are <0.05 while in figure 1A and 1B p values ranges from <0.05 to <0.0001.
RESPONSE 6: We are grateful to the reviewer, as this was a mistake. We have corrected the abstract to “MiR-146a-5p expression correlated inversely with the immune-related gene PTGS2, while its expression correlated directly with TSLP.” Regarding the p values, we have changed the specific p values in the text in lines 175, 179 and 180 in page5.
POINT 7
Figure 3: it should be of interest to present the results showing the individual value during bronchiolitis related to the individual value after disease remission in each case. How many of the 41 infants with bronchiolitis have the additional sample after disease remission?
RESPONSE 7:The reviewer has made an interesting observation and consequently we have represented all the individual data set points in a new Figure 3
Regarding how many infants with bronchiolitis have the additional sample after disease remission (post-bronchiolitis), only 12 subjects had this sample, having added this information in the manuscript: “The study population consisted of ten healthy controls, 41 infants diagnosed with bronchiolitis (of whom 12 infants had a post bronchiolitis sample), and 20 subjects with recurrent wheezing (ten with and ten without viral infection).” In lines 164-167 at page 4.
POINT 8
Figure 4: it is difficult to understand to which comparisons the asterisks refer.
RESPONSE 8: We regret the lack of understanding in this figure. All of the statistical comparisons were made for each treated group compared to the untreated control group. We have stated this in the figure legend at lines 234-235 in page 7.
POINT 9
Figure 5B: poly:IC stimulation increased TLR3 and TSLP expression but the effect in the wounded monolayer is the same as in the intact monolayer. There is no synergistic effects of poly:IC stimulation and epithelial damage.
RESPONSE 9: We agree with the reviewer on his comment, therefore we have added that “Nevertheless, unlike with miR-146a-5p expression, there was no synergistic effect on TLR3 and TSLP expression caused by poly:IC plus wounds.” As seen in lines 246-247 at page 8.
Round 2
Reviewer 2 Report
The response of the authors to my comments is satisfactory except for the response to my point 2.
In my opinion inclusion in the present manuscript of a clear definition of “recurrent wheezing” is mandatory. It is difficult to understand the meaning of the sentence “hospitalized infants with an episode of recurrent wheezing”. Children with “an episode” of wheezing are not with “recurrent” wheezing. This point should be clarified and a clear definition of recurrent wheezing should be included in the present manuscript. Moreover it should be specified whether in children defined “with recurrent wheezing” a diagnosis of bronchiolitis was ruled out
Author Response
Dear reviewer:
We understand the reviewer point of view on this subject. In order to rule out that misunderstandings are made while reading and comprehending the groups of study, we have re-labeled the recurrent wheezing group as wheezing episode in all the manuscript. Moreover, we have defined better the wheezing episode group in the manuscript saying that “ wheezing episode was defined as the presence of expiratory dyspnea and wheezing diagnosed by a physician in children with a previous episode of bronchiolitis, being the admission criteria the presence of hypoxia, apnea, respiratory distress and refusal of feeding. These patients were ruled out for bronchiolitis group, as these were no more their first episode of acute-onset expiratory dyspnea with previous signs of viral respiratory infection” As seen in lines 95 to 100 at pages 2-3 of the manuscript.